# Effect of a *Bacillus* Probiotic Compound on *Penaeus vannamei* Survival, Water Quality, and Microbial Communities

Xiaojuan Hu [1,2,3,4], Yu Xu [1], Haochang Su [1,2,3,4], Wujie Xu [1], Guoliang Wen [1], Chuangwen Xu [1], Keng Yang [1], Song Zhang [1] and Yucheng Cao [1,2,3,4,*]

1 South China Sea Fisheries Research Institute, Chinese Academy of Fishery Sciences, Guangzhou 510300, China
2 Key Laboratory of South China Sea Fishery Resources Exploitation & Utilization, Ministry of Agriculture and Rural Affairs, Guangzhou 510300, China
3 Guangdong Provincial Key Laboratory of Fishery Ecology and Environment, Guangzhou 510300, China
4 Southern Marine Science and Engineering Guangdong Laboratory (Zhuhai), Zhuhai 519000, China
* Correspondence: caoyucheng@scsfri.ac.cn or cyc_715@163.com; Tel.: +86-020-34063050

**Abstract:** Given the widespread use of commercial probiotics in aquaculture, it is important to evaluate the quality and environmental effects of these additives. Here, the effects of a *Bacillus* probiotic compound, BG4, on *Penaeus vannamei* survival rate, water quality factors, and microbial communities were assessed. An analysis of the BG4 powder confirmed the content of probiotic bacteria: 50.2% *Bacillus licheniformis*, 48.4% *Bacillus subtilis*, and 1.4% *Bacillus amyloliquefaciens*, consistent with the information on the product packaging. The effective bacterial quantity ($1.1 \times 10^9$ CFU/g) was higher than that indicated on the product label ($2 \times 10^8$ CFU/g). BG4 was added to a sterilized aquaculture water system, according to the indicated bacterial quantity; after 7 days, the maximum ammonia nitrogen, phosphate, and chemical oxygen demand degradation rates were 36.3%, 28.9%, and 15.2%, respectively. In the shrimp culture experiment, the survival rate of *P. vannamei* and water quality did not differ significantly between the BG4 and control groups. The bacterial quantity and high-throughput sequencing results indicated that *Bacillus* content in BG4 group decreased continuously from $3.5 \times 10^4$ CFU/mL to $6.9 \times 10^2$ CFU/mL. To ensure the desired effect of probiotics in commercial aquaculture applications, additional quality evaluations and scientific assessments are needed.

**Keywords:** commercial *Bacillus* probiotic; composition analysis; water quality factors; microbial community

**Key Contribution:** 1. The *Bacillus* probiotic, BG4, had a certain degradation effect on water quality factors in the sterilized aquaculture water system. 2. BG4 did not significantly influence water quality and the survival rate of *Penaeus vannamei* in 7 days in the shrimp culture system.

## 1. Introduction

Probiotics have been widely used in aquaculture to enhance the water quality of the aquaculture ponds [1]. Among them, *Bacillus* probiotics are most commonly used [2], with the goal of improving water quality [3], inhibiting harmful bacteria and algal blooms [4,5], and promoting the growth of fish and other farmed species [6]. Since the 1990s, commercial probiotics have been used in aquaculture in China. After 30 years of development, commercial *Bacillus* probiotics have been extensively applied in the production of the main aquaculture species, especially Pacific white shrimp, *Penaeus vannamei* [7], which is the most economically important crustacean species in global seafood aquaculture [8].

*Bacillus* is a Gram-positive, spore-producing, bacterium with a high level of stress tolerance [2]. Bacteria of this genus secrete extracellular enzymes that degrade organic matter, ammonia nitrogen ($NH_4^+-N$), and nitrite nitrogen ($NO_2^--N$) [9]. Liang et al. [10]

evaluated the pollutant degradation effect of *Bacillus licheniformis* BSK-4 in a system of cultured *Ctenopharyngodon idellus*. The authors reported that BSK-4 significantly degraded total nitrogen and $NO_2^- -N$, with maximum degradation rates of 35.95% and 96.70%, respectively. Lalloo et al. [11] showed that under the synergistic effect of *Bacillus subtilis* B001, *Bacillus cereus* B002, and *B. licheniformis* B003, the levels of $NH_4^+ -N$ and $NO_2^- -N$ in the water of aquaculture carp were significantly reduced, with degradation rates of 74% and 77%, respectively. Kewcharoen et al. [12] added *B. subtilis* AQAHBS001 to the feed of *P. vannamei* and found that its growth was significantly improved. Li et al. [13] demonstrated that the combination of *Bacillus giant* AO and *B. subtilis* BS significantly reduced the level of $NH_4^+ -N$ and $NO_2^- -N$ in the aquaculture water of crucian carp, increased the microbial diversity in aquaculture water, and reduced the abundance of pathogenic bacterial genera. During production, commercially circulated *Bacillus* products prepared from *Bacillus* bacterial fluid undergo multiple processes, such as industrialized fermentation, product circulation, and storage [14]. Thus, the application effect of inoculant products and the characteristics of the pure bacterial strains may vary [15]. Li and Boyd [16] studied the degradation effects of 12 types of bacterial agents on total ammonia nitrogen and $NO_2^- -N$ in an environmentally controlled aquarium and found no obvious degradation effect.

Researchers have focused on the isolation and screening of high-efficiency *Bacillus* strains and their effects [17]; the composition of commercial *Bacillus* products and their effect on water microbial communities and water quality are not well understood [18,19]. However, in order to standardize the management of fishery probiotics, scientific evaluation of the effect of current probiotics (such as commercial *Bacillus* probiotics) is necessary and would aid in the development of evidence-based proposals for sustainable development of the industry.

The commercial probiotic *Bacillus* compound, BG4, a common aquatic probiotic product in Chinese fisheries, was selected as the research object. We analyzed the quantity and composition of *Bacillus* in BG4 and the effect of BG4, in water sourced from an aquaculture pond and then sterilized, on the *P. vannamei* survival rate and microbial community structure. The results contribute to a more comprehensive evaluation of the quality of commercial *Bacillus* products in aquaculture and the standardized management of probiotics applied in aquaculture, with the goal of promoting sustainable development of fishery microbial resources.

## 2. Materials and Methods

### 2.1. Experimental Material

The commercial compound *Bacillus* probiotic BG4 was collected from a aquafarm in Guangdong province of China, which was a common aquatic probiotic product.

### 2.2. Bacterial Composition Analysis of BG4

#### 2.2.1. Activation of BG4

According to the method of activation of BG4 indicated on the product label, 2.5 g of BG4 and 1.5 g of brown sugar were added to an Erlenmeyer flask containing 500 mL of sterile water and placed in a shaker for 12 h at a culture temperature of 30 °C and a rotation speed of 180 rpm. Three parallel samples were prepared.

#### 2.2.2. Bacterial Composition Analysis of BG4

BG4 (25 g) was added to 225 mL of 0.85% sterilized normal saline and homogenized to make a suspension. Subsequently, 0.85% sterilized normal saline was used for gradient dilution, and three appropriate dilutions (−5, −6, and −7 gradients) were chosen. The bacterial suspension was heated in a water bath at 80 °C ± 1 °C for 10 min. Thereafter, 0.1 mL of the suspension was spread on nutrient agar plates and incubated at 37 °C for 48 h [20]. After colony formation, the number of colonies was counted. The number of *Bacillus* species in 1 g of BG4 was counted.

### 2.2.3. Identification of *Bacillus* Species

The bacteria cultured in the nutrient agar plates were isolated and purified. And single colonies were separated and purified on the basis of the morphological characteristics of the colonies on the counting plate. Single colonies were selected, and bacterial DNA was extracted using TIANamp Bacteria DNA Kit (Tiangen Biotech, Beijing, China). The 16S rDNA fragment was amplified via PCR with the primers 8F (5'-AGAGTTTGATCCTGGCTCAG-3') and 1492R (5'-GGTTACCTTGTTACGACTT-3'). In a 50-μL reaction system, 2 μL of template DNA was added. The reaction conditions were as follows: 95 °C for 4 min, 95 °C for 1 min, 48 °C for 1 min, and 72 °C for 2 min, 30 cycles; and 72 °C for 10 min. The amplified products were detected using 1.0% agarose gel electrophoresis and sent to Sangon Biotech (Shanghai) Co., Ltd. (Shanghai, China) for sequencing.

### 2.3. Analysis of the Effect of BG4 on the Quality of Sterilized Aquaculture Water

#### 2.3.1. Preparation of Feed Solutions

A commercially available compounded feed (Yuehai Feed Group Co., Ltd., Zhanjiang, China) containing 41% crude protein, 5% crude fat, and 16% ash as the main ingredients (mass fraction) was used in this study. The feed was dried and crushed, and 100 g of the crushed feed was mixed with 1 L of purified water. The feed solution was stored at 25 °C for 24 h.

#### 2.3.2. Preparation of Sterilized Aquaculture Water

The feed solution was mixed with aquaculture water from the intensive aquaculture pond, at a salinity of 10‰ and a volume ratio of 1:100. The initial water quality concentrations were adjusted using sodium nitrite and ammonium chloride. After the adjustment, the initial chemical oxygen demand (COD) concentration of the aquaculture water was 10–15 mg/L, the initial concentrations of $NH_4^+-N$ and $NO_2^--N$ were 1–3 mg/L, and the initial concentration of phosphate ($PO_4^{3-}-P$) was 1–2 mg/L. These values were within the concentration range in water in typical aquaculture ponds [21]. Subsequently, the solution was aliquoted from 600 mL to 1 L Erlenmeyer flasks and sterilized at 121 °C for 20 min.

#### 2.3.3. Treatment Groups

The BG4 and control groups were set up as three parallel replicates; in the BG4 group, BG4 was added to the sterilized aquaculture water according to the specified bacterial quantity ($5 \times 10^4$ CFU/mL) in the product description. In the control group, normal saline was added to the same volume as the treatment mixture used in the BG4.

#### 2.3.4. Water Quality Indicators

Water was sampled on experimental days 0, 3, and 7. The $NH_4^+$-N, $NO_2^-$-N, nitrate nitrogen ($NO_3^--N$), $PO_4^{3-}-P$, and COD concentrations were measured using indophenol blue spectrophotometry, naphthyl ethylenediamine spectrophotometry, zinc cadmium reduction method, phosphomolybdate blue spectrophotometry, and alkalic potassium permanganate according to GB17378.4−2007 [22].

### 2.4. Effects of BG4 on Shrimp and Aquatic Environment

The effects of the bacterial cultures contained in BG4 on *P. vannamei* and the aquatic environment were investigated.

#### 2.4.1. Experimental System

Similar sized (PL10) cultured *P. vannamei* were selected. The shrimp were allowed to adapt to the aquaculture environment for 1 week, after which they were randomly placed in an aquarium with a size of 50 cm × 30 cm × 80 cm. Six aquariums (30 L each) were set up, and each aquarium contained 30 shrimp. After allowing the shrimp to adapt to the aquarium environment, they were fed twice a day, at 8:00 and 18:00, until most of them no longer fed. The experimental water came from an intensively farmed shrimp pond. During the experiment, the water was aerated without changing the water.

2.4.2. Experimental Design

BG4 and control groups were prepared with three parallel setups. In the BG4 group, BG4 was added to aquaculture water where *P. vannamei* were cultured in an amount based on the bacterial quantity indicated in the product description ($5 \times 10^4$ CFU/mL). In the control group, the same volume of normal saline as BG4 was added.

2.4.3. Detection of Indicators

- Survival rate of *Penaeus vannamei*

The survival status of *P. vannamei* was observed in the aquarium on experimental days 0, 1, 3, and 7, and the number of shrimp in the aquarium was recorded. Any dead shrimp were immediately removed, using a fresh dip net each time. The survival rates were calculated.

- Water quality

On experimental days 0, 1, 3, and 7, the basic physical and chemical indicators of water quality, namely temperature, pH, and dissolved oxygen (DO), as well as the concentration of $NH_4^+-N$, $NO_2^--N$, $NO_3^--N$, $PO_4^{3-}-P$, and COD, were measured from the water samples from the aquariums. Concentrations were measured as previously described.

- Bacterial quantity and microbial community structure

From the water samples collected on days 0, 1, 3, and 7, appropriate dilutions of the bacterial solutions were aspirated using sterile pipettes. The number of bacteria in the water sample was counted with a hemocytometer. Heterotrophic bacteria were quantified using the agar plate counting method and counted after 48 h of culture at a constant temperature of 30 °C.

Water samples (100 mL) from the BG4 and control groups were filtered through a 0.22 μm filter membrane (Millipore, Boston, USA). Total bacterial DNA was extracted from water samples using a microbial DNA extraction kit (Omega, Norcross, USA). The primers 515F (5′-GTGCCAGCMGCCGCGGTAA-3) and 907R (5′-CCGTCAATTCMTTT RAGTTT-3′) were used to amplify the 16S V4-V5 variable region, and an amplified library was constructed. The composition of the microbial community in the water of each experimental group was analyzed by comparing it with that in the SILVA database.

*2.5. Statistical Analysis*

The degradation rate (*R*) of each water quality indicator was calculated as follows:

$$R(\%) = \frac{(initial\ concentration\ C_0 - concentration\ C_t\ after\ the\ experiment)}{initial\ concentration\ C_0} \times 100$$

All data are presented as the mean ± standard deviation. Significant differences in the data of each group were compared through one-way ANOVA after LSD test using SPSS software (version 20.0); the significance level was set at $p < 0.05$.

**3. Results**

*3.1. Analysis of Bacterial Species Composition of BG4*

The BG4 product was found to contain more than $2 \times 10^8$ CFU/g *Bacillus*, which is the main ingredient marked on its label. The effective bacterial quantity of BG4 obtained by spread plate method was $1.1 \times 10^9$ CFU/g. The single colonies were separated and purified on the basis of the morphological characteristics of the colonies on the counting plate. The single colonies were identified individually using 16S rDNA. The results showed that the probiotic BG4 product was composed of 50.2% *B. licheniformis*, 48.4% *B. subtilis*, and 1.4% *B. amyloliquefaciens* (Table 1), which was consistent with the main *Bacillus* components indicated on the product packaging. The bacterial count was higher than the count that was indicated on the product packaging ($2 \times 10^8$ CFU/g).

**Table 1.** Bacterial species composition of commercial *Bacillus* probiotic BG4.

| Bacterial Species Composition | *Bacilluslicheniformis* | *Bacillus subtilis* | *Bacillus amyloliquefaciens* |
|---|---|---|---|
| Bacterial quantity (CFU/g) | $5.52 \times 10^8$ | $5.32 \times 10^8$ | $1.54 \times 10^7$ |
| Percentage of quantity | 50.2% | 48.4% | 1.4% |

*3.2. Effect of BG4 on the Quality of Sterilized Aquaculture Water*

The COD in the BG4 group on day 7 was 15.2%, decreasing from 15.44 to 13.09 mg/L ($p < 0.05$, Figure 1A). The $NO_2^- - N$ concentration decreased from 1.12 to 0.72 mg/L, and the degradation rate on day 7 was 36.3% ($p < 0.05$, Figure 1B). The $NO_3^- - N$ degradation rate was 17.4% compared with that of the control group on day 7 (Figure 1D). The $PO_4^{3-} - P$ concentration decreased from 1.51 mg/L to 1.09 and 1.17 mg/L on days 3 and 7, respectively, with a degradation rate of 28.9% and 22.5%, respectively ($p < 0.05$, Figure 1E). The quantity of bacteria in the BG4 group increased from $2 \times 10^4$ CFU/mL on day 0 to $1 \times 10^8$ CFU/mL on days 3 and 7 (Figure 1F).

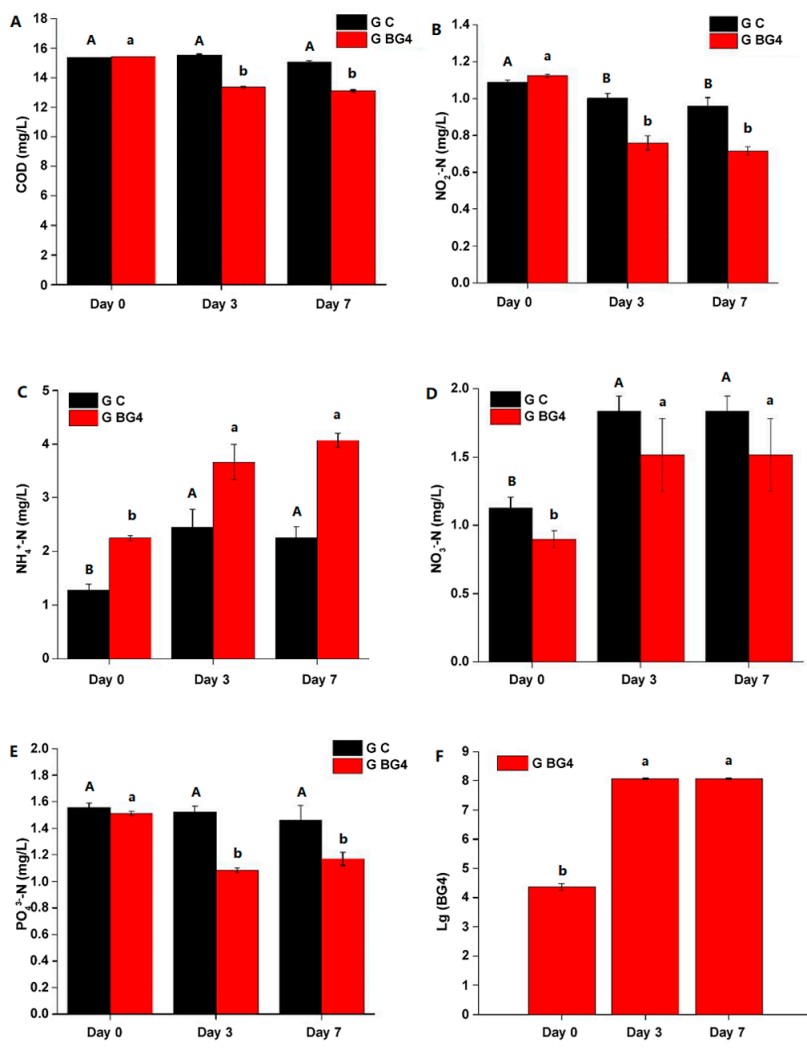

**Figure 1.** Effect of commercial *Bacillus* probiotic BG4 on the water quality indicators of sterilized aquaculture water and variation in bacterial quantity. (**A**) Chemical oxygen demand (COD), (**B**) $NO_2^- - N$, (**C**) $NH_4^+ - N$, (**D**) $NO_3^- - N$, (**E**) $PO_4^{3-} - P$, and (**F**) Bacterial quantity. Different uppercase letters indicate significant differences in data at different time points between control groups ($p < 0.05$), and different lowercase letters indicate significant differences in data at different time points between BG4 groups ($p < 0.05$).

*3.3. Effects of BG4 on Aquaculture Organisms and Aquatic Environment*

3.3.1. Survival rate of *Penaeus vannamei*

The survival rates of *P. vannamei* was greater than 94% in both the BG4 and control groups; there was no significant difference between BG4 treatment and control on days 0, 1, 3, and 7 ($p > 0.05$; Figure 2A).

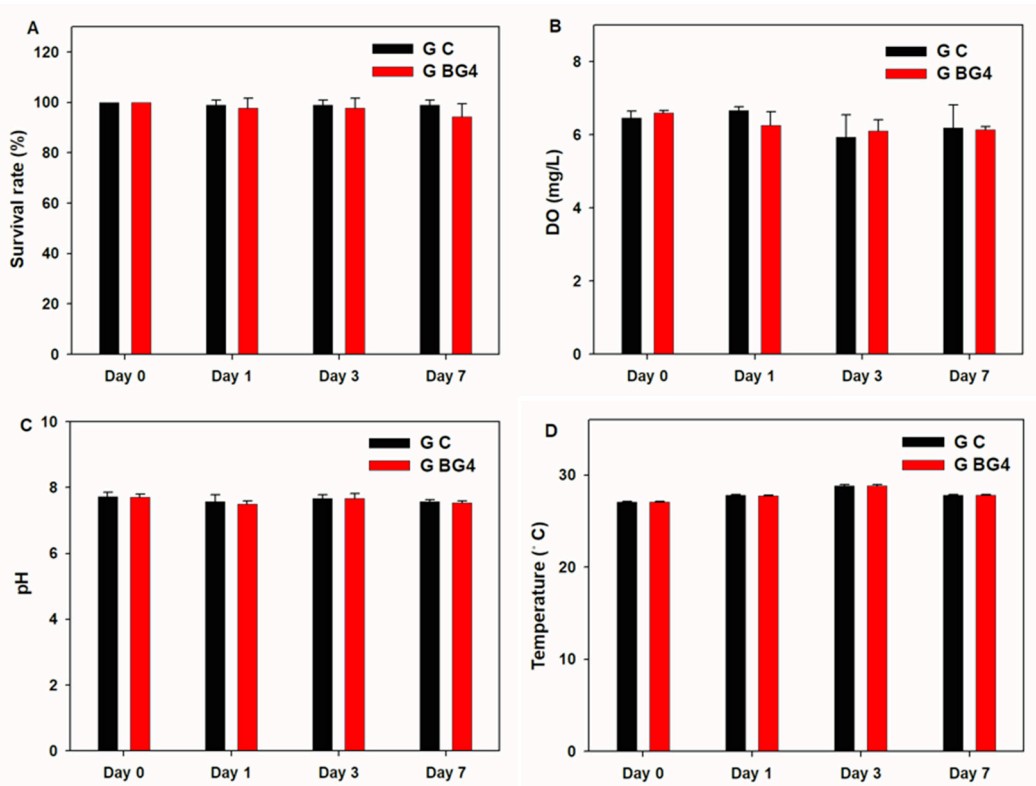

**Figure 2.** Survival rate of *Penaeus vannamei* (**A**), dissolved oxygen (**B**), pH (**C**), and temperature (**D**) of aquaculture water in the BG4 and control groups.

3.3.2. Effects of BG4 on Aquatic Environment

- Basic indicators

The DO, pH, and temperature of the culture water in the control and BG4 groups were within the optimal ranges for *P. vannamei* and did not differ significantly ($p > 0.05$). The concentration of DO in the control and BG4 groups was 5.94–6.66 mg/L and 6.10–6.59 mg/L, respectively (Figure 2B). The pH in the control and BG4 groups was 7.57–7.73 and 7.50–6.59, respectively (Figure 2C). The temperature range of the two groups was 27.03–28.83 °C (Figure 2D).

- Water quality

The concentrations of N and P forms in the water of both BG4 and control groups showed an upward trend, which could be attributed to the addition of feed over the course of the 7 days. During the experiment, the $NH_4^+-N$ (Figure 3A), $NO_2^--N$ (Figure 3B), $NO_3^--N$ (Figure 3C), $PO_4^{3-}-P$ (Figure 3D), and COD (Figure 3E) concentrations did not significantly differ between the two groups ($p > 0.05$).

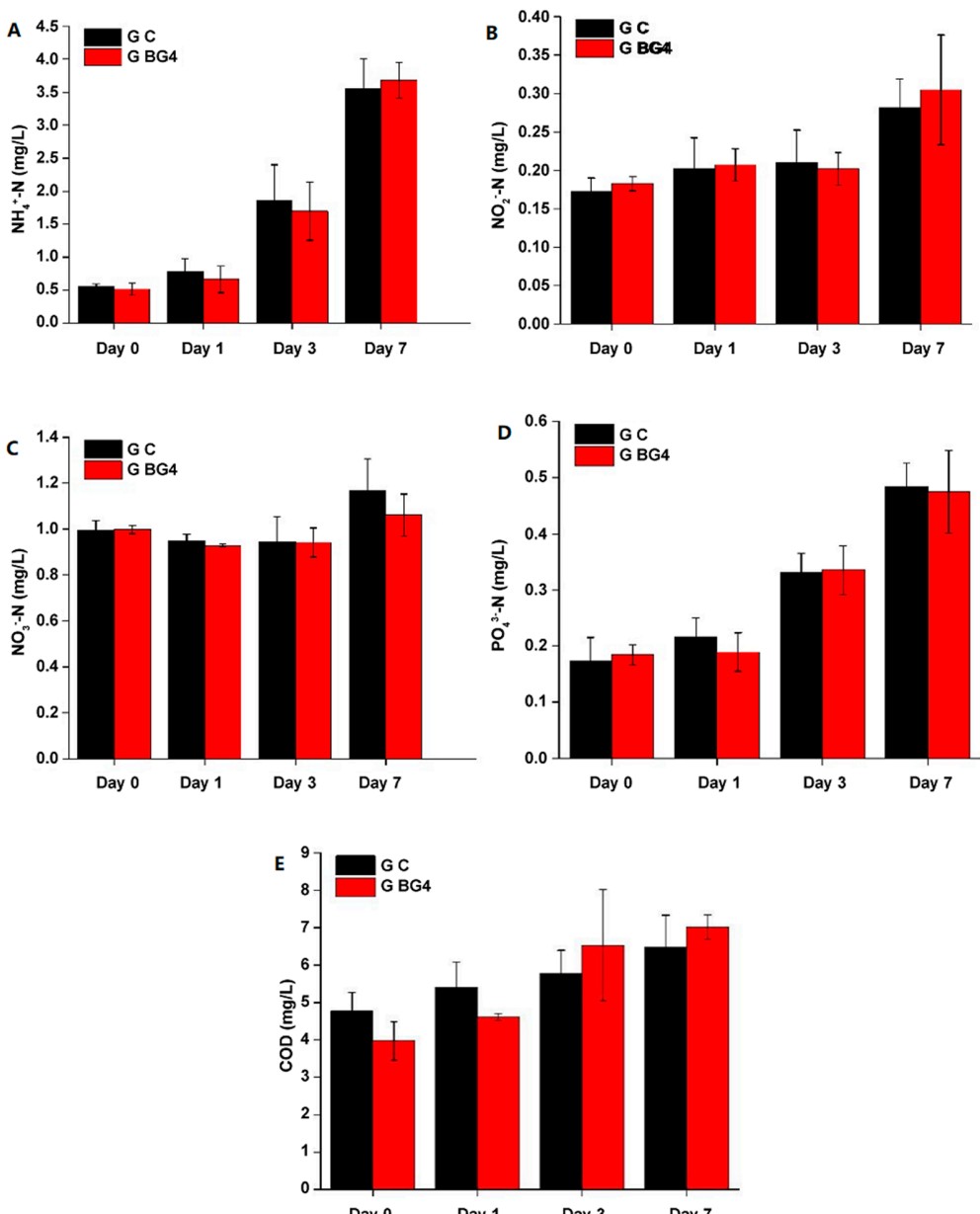

**Figure 3.** Effect of commercial *Bacillus* probiotic BG4 on the water quality indicators in the shrimp culture water. (**A**) $NH_4^+-N$, (**B**) $NO_2^--N$, (**C**) $NO_3^--N$, (**D**) $PO_4^{3-}-P$, and (**E**) COD.

### 3.4. Effects of BG4 on the Number of Bacteria and Community Structure in the Shrimp Culture Water

3.4.1. Changes in the Total Number of Bacteria in the Shrimp Culture Water

Throughout the experiment, the total number of bacteria in the water remained stable at $3 \times 10^6$–$4 \times 10^6$ cells/mL, and no significant differences were found between the groups ($p > 0.05$; Figure 4). The application of the inoculant product in accordance with its recommended amount only slightly affected the total number of bacteria in the water.

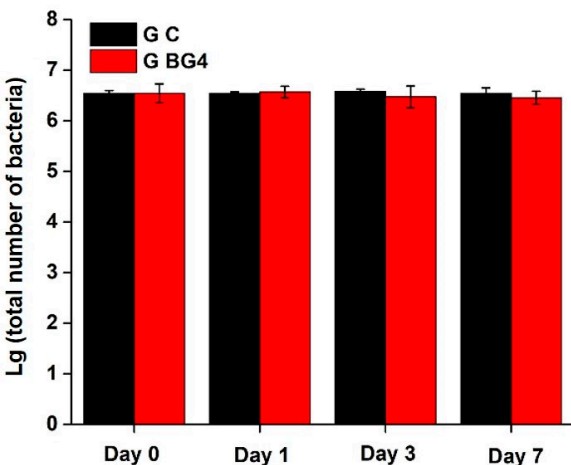

**Figure 4.** Variation in bacterial quantity of shrimp culture water of the BG4 and control groups.

3.4.2. Changes in Bacterial Community Structure in the Shrimp Culture Water

Changes in the composition of the bacterial communities in the BG4 and control groups during the experiment were analyzed using high-throughput sequencing. At the beginning of the experiment, the dominant bacteria in the two groups were members of the phyla Proteobacteria, Bacteroidetes, and Firmicutes, which, in the control group, represented 50.48%, 21.98%, and 17.44%, respectively, of the bacteria. In the BG4 group, the relative abundance of these phyla were 44.46%, 18.19%, and 26.60%, respectively (Figure 5). On experimental day 1, the proportion of bacteria in Proteobacteria, Bacteroides, and Firmicutes was 30.11%, 40.41%, and 20.36% in the control group and 49.02%, 26.62%, and 15.45% in the BG4 group, respectively. On day 3, the corresponding proportions were 51.50%, 38.59%, and 5.25% in the control group and 64.58%, 24.83%, and 0.47% in the BG4 group. In the BG4 group, the relative abundance of Firmicutes was lower than that in the control group ($p < 0.05$). On day 7, Proteobacteria, Bacteroides, and Firmicutes represented 55.88%, 27.68%, and 6.40% of all bacteria in the control group and 54.75%, 31.76%, and 0.65% in the BG4 group. The percentage of bacteria in the BG4 group that were members of phylum Firmicutes was less than the corresponding portion in the control group ($p < 0.05$).

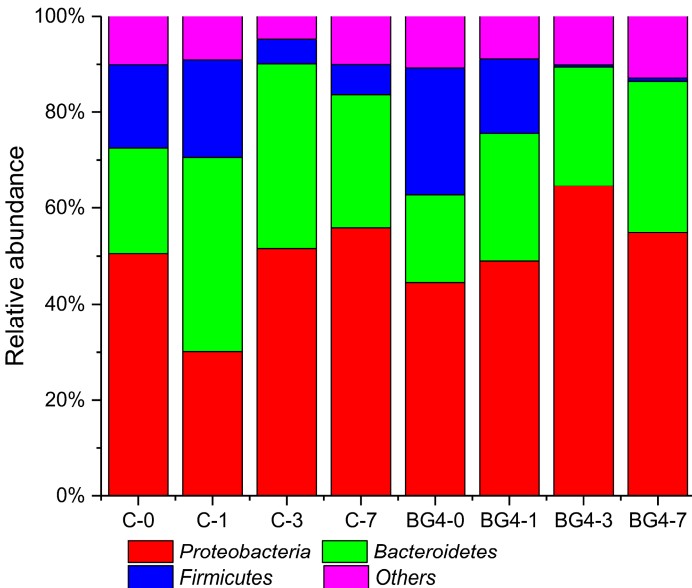

**Figure 5.** Relative abundance of bacterial communities in the shrimp culture water of the BG4 and control groups at the phylum level.

Twelve dominant bacterial genera were detected: *Bacillus, Bergeyella, Neisseria, Acidovorax, Bacteroides, Acinetobacter, Moraxella, Flavobacterium, Erythrobacter, Gemmobacter, Lactobacillus*, and *Rhodopseudomonas*. On day 0, the average proportion of *Bacillus* was 1.01% in the BG4 group (Figure 6), which was significantly higher than that in the control group ($p < 0.05$). The average proportion of *Bergeyella* was significantly higher in the BG4 group than in the control group ($p < 0.05$), and the average proportions of *Moraxella* and *Flavobacterium* in the BG4 group were significantly lower than those in the control group ($p < 0.05$). On day 1, the average proportion of *Bacillus* was 0.21% in the BG4 group (Figure 6), which was not significantly different from that in the control group ($p < 0.05$). In the BG4 group, the relative abundance of *Acidovorax, Erythrobacter, Flavobacterium*, and *Gemmobacter* was significantly higher than the corresponding values in the control group ($p < 0.05$). On day 3, the average proportion of *Bacillus* was 0.07% in the BG4 group (Figure 6), which was significantly lower than that in the control group ($p < 0.05$). The relative abundance of both *Flavobacterium* and *Gemmobacter* in the BG4 group was significantly higher than that in the control group ($p < 0.05$), and the average proportion of *Bergeyella* was significantly lower in the BG4 group than in the control group ($p < 0.05$). On day 7, the average proportion of *Bacillus* was 0.02% in the BG4 group (Figure 6), which was significantly lower than that in the control group ($p < 0.05$). Similarly, the relative abundance of *Acinetobacter* was significantly lower in the BG4 group than in the control group ($p < 0.05$); however, the average proportion of *Flavobacterium* was significantly higher in the BG4 group than in the control group ($p < 0.05$).

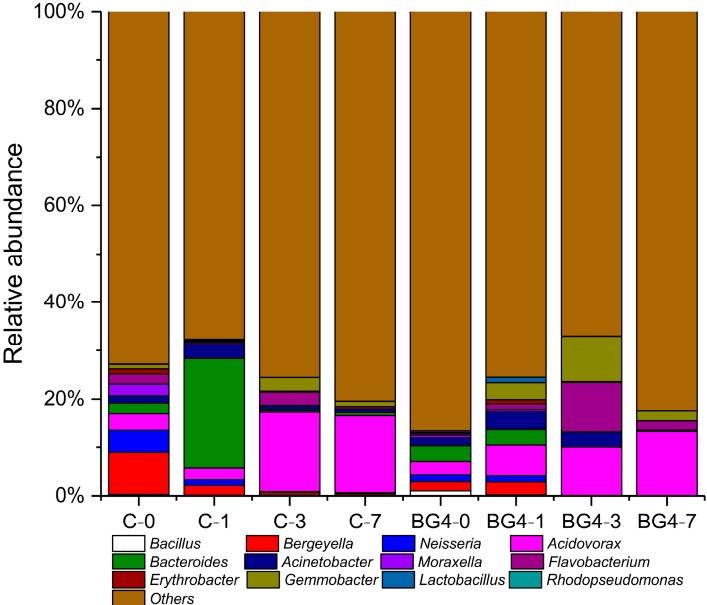

**Figure 6.** Relative abundance (genus level) of bacterial communities in the shrimp culture water of the BG4 and control groups.

Based on the number of bacteria in the shrimp culture water and the high-throughput sequencing results on experimental days 0, 1, 3, and 7, the relative abundance of *Bacillus* at the genus level was 1.01%, 0.21%, 0.07%, and 0.02%, respectively; and the corresponding counts were $3.5 \times 10^4$, $7.3 \times 10^3$, $2.4 \times 10^3$, and $6.9 \times 10^2$ CFU/mL. The initial quantity of *Bacillus* added in the BG4 group was $3.5 \times 10^4$ CFU/mL, which was the same as that in the product description (Figure 7).

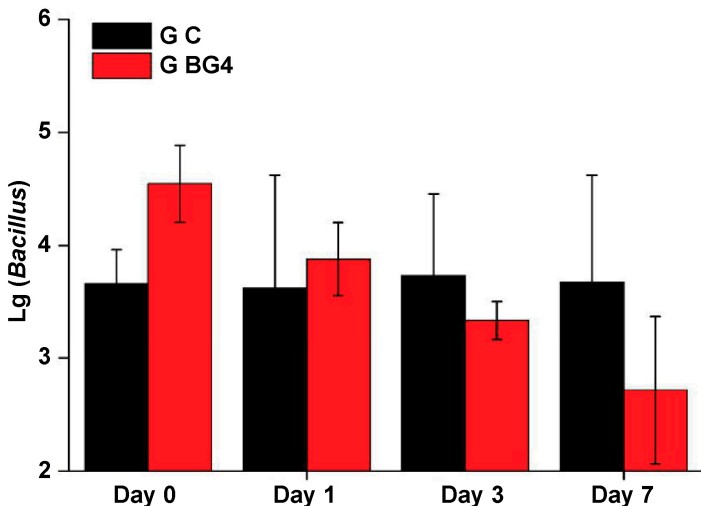

**Figure 7.** Changes in the bacterial count of *Bacillus* in the control and BG4 groups of shrimp culture water over 7 days.

## 4. Discussion

Since the 1990s, commercial probiotics have been used in aquaculture in China. Species of *Bacillus* produce a large number of highly active proteases, lipases, amylases, and other enzymes, which rapidly degrade macromolecular organic compounds, such as proteins, fats, and starch, present in aquaculture water [19,23]. Commercial *Bacillus* probiotics have been extensively applied in the culture of several aquaculture species, especially shrimp [24–26]. According to the Catalog of Feed Additives of the Ministry of Agriculture of China 2013, Announcement No. 2045, *Bacillus* species that can be used as additives in aquaculture include *B. licheniformis*, *B. subtilis*, *B. pumilus*, *B. coagulans*, and *B. lentus*. In this study, the main components of BG4 were *B. licheniformis* and *B. subtilis*, which meet the requirements of the Catalog of Feed Additives. In addition, *B. licheniformis* and *B. subtilis* are the most studied species of *Bacillus* with respect to purification function [27,28]. The quantity of *Bacillus* in the product used in the present study exceeded the quantity marked on the product package, meeting the requirements of product quality.

In studies of the aquaculture effects of *Bacillus*, the results show that *Bacillus* species promote effective aquaculture by improving and regulating water quality, microalgae communities [14], and microbial communities [5]. Among them, *B. licheniformis* and *B. subtilis* are often considered to have good water purification function. Cao et al. [27] studied the application effect of *B. licheniformis* in the culture of *Ctenopharyngodon idellus* and found that the concentration of $NH_3-N$ and $NO_2^--N$ in the *Bacillus* group decreased by 74.6% and 69.3%, respectively. Liu et al. [28] found that *B. subtilis* WH1 significantly reduced $NO_2^--N$, $NH_3-N$, and COD concentrations in water ($p < 0.05$). The addition of *B. subtilis* ($10^8$ CFU/mL) directly to the rearing water maintained the concentrations of $NH_4^+-N$, $NO_2^--N$, and $NO_3^--N$ within the tolerable ranges for shrimp culture [29]. *Bacillus* bacteria decompose organic matter in water, reducing COD in water and converting organic matter into inorganic nutrients such as $NH_4^+-N$ [30]. These results indicate that the *Bacillus* probiotics of BG4 has a certain effect on water purification.

The goal of the present study in a sterilization system was to evaluate the effect of probiotics on water quality without the interference of other factors. The results of the present study are consistent with previous reports, to a certain extent. In the sterilized aquaculture water system, the maximum $NO_2^--N$, $PO_4^{3-}-P$, and COD degradation rates in the BG4 group were 36.3%, 28.9%, and 15.2%, respectively, and their concentrations decreased from 1.12, 1.51, and 15.44 mg/L to 0.72, 1.09, and 13.09 mg/L, respectively. However, the water quality did not significantly differ between the BG4 and control groups over 7 days. This was consistent with the research of Li and Boyd [16]. The authors

found that, in an environmentally controlled aquarium, 12 bacterial agents showed no obvious effect on water quality. Likewise, Zheng et al. [15] studied the effect of a *Bacillus* preparation on the mixed culture water of mussels and fish and found that the probiotic had no significant effect on the level of nitrogen and phosphorus in the water. It is worth noting that in an environmentally controlled aquarium, the quantity of microalgae is very small; therefore, the normal synergistic effect of bacteria and microalgae in aquatic environments cannot be realized in the sterilized water system.

As the experiment progressed, the bacterial community structure of the control and BG4 groups changed significantly, possibly because regular addition of feed changed the nutritional status of the water [31]. The composition of bacterial community structure in the control and BG4 groups was different. This result was consistent with related reports, showing that commercial probiotic products can change the natural bacterial community [25]. The effect of commercial probiotic products on natural bacteria may be related to the competition between foreign and natural bacteria or the adaptation of foreign bacteria to the natural environment [32]. The number of bacteria in shrimp culture water and the high-throughput sequencing results indicated that the bacterial count of *Bacillus* in the BG4 group decreased continuously from $3.5 \times 10^4$ CFU/mL to $6.9 \times 10^2$ CFU/mL during the experiment. This indicated that *Bacillus* did not proliferate in this experimental system, and its function may be affected by the shrimp culture water environment [33]. Li et al. [34] and Cao et al. [35] studied cultured ponds of *P. vannamei* and the mixed ponds of shrimp–fish–crab and found that the quantity of *Bacillus* was between 60 and $1.7 \times 10^3$ CFU/mL. Compared with this data, it was found that the bacterial count of *Bacillus* decreased to the level of the common aquaculture pond after 7 days. This above research showed that the characteristics and effects of probiotic products may differ from those of pure bacterial strains.

## 5. Conclusions

We conclude that the *Bacillus* probiotic, BG4, has a certain degradation effect on water quality factors such as ammonia nitrogen, phosphate, and COD in the sterilized aquaculture water system, and their degradation rates were 36.3%, 28.9%, and 15.2%, respectively. However, the survival rate of *P. vannamei* and water quality did not differ significantly between the BG4 and control groups in the shrimp culture experiment. Given that the difficulties of modelling the pond conditions in laboratory conditions and the specific objectives of probiotic treatment vary, as do the ecological characteristics of the different strains, it is necessary to carry out the quality evaluation and scientific assessments of probiotics at the level of the experimental system as well as aquaculture systems.

**Author Contributions:** Conceptualization, Y.C.; methodology, Y.C. and K.Y.; formal analysis, X.H. and H.S.; investigation, S.Z. and C.X.; data curation, X.H. and W.X.; writing—original draft preparation, X.H.; writing—review and editing, X.H. and Y.X.; Supervision, Y.C. and G.W.; funding acquisition, X.H., Y.C. and G.W. All authors have read and agreed to the published version of the manuscript.

**Funding:** This research was funded by the key R&D Program of Guangdong Province (2021B0202040001); the Central Public-Interest Scientific Institution Basal Research Fund, CAFS (2020TD54); the Guangdong Provincial Special Fund For Modern Agriculture Industry Technology Innovation Teams (2022KJ149); the earmarked fund for CARS-48; and the Central Public-Interest Scientific Institution Basal Research Fund, South China Sea Fisheries Research Institute, CAFS (2021SD08).

**Institutional Review Board Statement:** The animal study protocol was approved by the Animal Care and Use Committee of the South China Sea Fisheries Research Institute, Chinese Academy of Fishery Sciences (SCSFRI-CAFS, No. nhdf2022-02).

**Data Availability Statement:** The datasets generated and analyzed during the current study are available from the corresponding author on reasonable request.

**Conflicts of Interest:** The authors declare no conflict of interest.

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
