# Peer review of "Effect of a Bacillus Probiotic Compound on Penaeus vannamei Survival, Water Quality, and Microbial Communities"

_fishes, doi:10.3390/fishes8070362_

Round 1

Reviewer 1 Report (Previous Reviewer 1)

The MS has improved a lot. Now it is acceptable for publication fulfilling all expected requirements for scientific papers. However, I have to mention that, according to a quick search made by Elicit, the product investigated very thoroughly seems to be totally unknown and unavailable internationally which may diminish its value.

Author Response

Response to Reviewer 1 Comments:

The MS has improved a lot. Now it is acceptable for publication fulfilling all expected requirements for scientific papers. However, I have to mention that, according to a quick search made by Elicit, the product investigated very thoroughly seems to be totally unknown and unavailable internationally which may diminish its value.

Response: Thank the reviewer for the affirmation of this manuscript. This is how we think about the product information of a Bacillus probiotic compound. Considering the repeatability of scientific research, we are well aware that experimental reagents need to disclose product source information such as manufacturers. However, in this study, a "double blind" evaluation method was adopted, selecting a common Bacillus probiotic in the Chinese fishing probiotics market for evaluation and research, in order to objectively evaluate the probiotics and avoid suspicion of commercial interests.

In the revised manuscript, “The commercial compound Bacillus probiotic BG4 was sourced from a well-known aquatic probiotic production company in China.” was revised to “The commercial compound Bacillus probiotic BG4 was collected from a aquafarm in Guangdong province of China, which was a common aquatic probiotic product.” in line 85-86.

Reviewer 2 Report (Previous Reviewer 2)

The authors have done an excellent job of revising the manuscript per reviewer’s comments. I have suggested some additional minor edits in verb tense and word usage for the authors to consider. When these are corrected, the manuscript can be accepted for publication.

Acceptable

Author Response

Response to Reviewer 2 Comments:

The authors have done an excellent job of revising the manuscript per reviewer’s comments. I have suggested some additional minor edits in verb tense and word usage for the authors to consider. When these are corrected, the manuscript can be accepted for publication.

Response: Thank the reviewer for the affirmation of this manuscript. According to your opinions, We made point-to-point revisions.

Point 1- line 32: “has” Replace with "had".

Response 1: Thank you very much for your comments. In the revised manuscript, “has” was revised to “had” in line 32.

Point 2 - line 34: “and” Replace with "or".

Response 2: Thank you very much for your comments. According to the results of this manuscript, the Bacillus probiotic BG4 did not significantly influence water quality and the survival rate of Penaeus vannamei in 7 days in the shrimp culture system. So we think use “and” more properly.

Point 3 - line 85-86: Source should be identified by name and location.

Response 3: Thank you very much for your comments. This is how we think about the product information of a Bacillus probiotic compound. Considering the repeatability of scientific research, we are well aware that experimental reagents need to disclose product source information such as manufacturers. However, in this study, a "double blind" evaluation method was adopted, selecting a common Bacillus probiotic in the Chinese fishing probiotics market for evaluation and research, in order to objectively evaluate the probiotics and avoid suspicion of commercial interests.

In the revised manuscript, “The commercial compound Bacillus probiotic BG4 was sourced from a well-known aquatic probiotic production company in China.” was revised to “The commercial compound Bacillus probiotic BG4 was collected from a aquafarm in Guangdong province of China, which was a common aquatic probiotic product.” in line 85-86.

Point 4 - line 102: The word "the" was deleted.

Response 4: Thank you very much for your comments. In the revised manuscript, "the" was deleted in line 102.

Point 5 - line 115: The word "compound" Replace with "compounded".

Response 5: Thank you very much for your comments. In the revised manuscript, “compound” was revised to “compounded” in line 115.

Point 6 - line 150: The word "inflated" is an incorrect term. Please replace with "aerated".

Response 6: Thank you very much for your comments. In the revised manuscript, “inflated” was revised to “aerated” in line 151.

Point 7 - line 169: Remove one of the periods.

Response 7: Thank you very much for your comments. In the revised manuscript, one of the periods was removed in line 169.

Point 8 - line 186: Remove one of the periods.

Response 8: Thank you very much for your comments. In the revised manuscript, one of the periods was removed in line 188.

Point 9 - line 211: The word "indicated" Replace with "indicate".

Response 9: Thank you very much for your comments. In the revised manuscript, the word "indicated" was revised to "indicate" in line 214.

Point 10 - line 334: Redundant, replace with "The".

Response 10: Thank you very much for your comments. In the revised manuscript, “In the present study, the goal of the present study in a sterilization system was to evaluate the effect of probiotics on water quality without the interference of other factors. ” was revised to “The goal of the present study in a sterilization system was to evaluate the effect of probiotics on water quality without the interference of other factors. ” in line 340.

Point 11 - line 368: First statement is not necessary, replace with "we". And the word “Bacillus”needs to be italicized.

Response 11: Thank you very much for your comments. In the revised manuscript, “On the basis of the above analysis, we conclude that the Bacillus probiotic BG4, has a certain degradation effect on water quality factors such as ammonia nitrogen, phosphate, and COD in the sterilized aquaculture water system, which degradation rates were 36.3%, 28.9%, and 15.2%,respectively.” was revised to “We conclude that the Bacillus probiotic BG4, has a certain degradation effect on water quality factors such as ammonia nitrogen, phosphate, and COD in the sterilized aquaculture water system, which degradation rates were 36.3%, 28.9%, and 15.2%, respectively.” in line 374- 377.

Point 12 - line 334: The word “P. vannamei” needs to be italicized.

Response 12: Thank you very much for your comments. In the revised manuscript, “However, the survival rate of P. vannamei and water quality did not differ significantly between the BG4 and control groups in the shrimp culture experiment.” was revised to “However, the survival rate of P. vannamei and water quality did not differ significantly between the BG4 and control groups in the shrimp culture experiment.” in line 377.

Reviewer 3 Report (Previous Reviewer 3)

still the authors did not address most of the comments submitted before. 

Materials section

1- line 86: please mention the name of the company and all other details between brackets.

2- line 86: how do you think the difference in bacterial concentration written on the label from the actual concentration will affect the shrimp?

3- topic 2.2.1.: activation of BG4, please include your reference for this method or if it is lab-induced method you can mention this as well.

4- before you add the identification step, you need to explain the culture of your bacteria. There is a missing part in your manuscript, you need to explain how you prepared bacterial culture and grown your bacteria then come to the isolation and identification step.

5- line 149: "During the experiment, the gas was 149 inflated without changing the water." You should re-write this part as it is oxygen supply not gas.

6- you repeated the experimental design twice, as under topic 2.3.3. treatment groups and topic 2.4.2. experimental design. 

The same for topic 2.3.4. and topic 2.4.3. (line 168) you repeated the same water quality measurements twice.

Again it is the same situation for " Bacterial quantity and microbial community structure " you are repeating the same paragraphs and it is not professional.

7- line 163: do not use "aquaculture organism" use the species name in your experiment.

8- statistical analysis: the equation regarding "R" should not be placed here, instead it should be placed under water quality.

9- figure1: how do you explain the differences between your treatment group (BG4) and control group in day 0? Is it logic? If there are differences in day zero, so there is something wrong with your experimental design and the downstream results are no more reliable.  

10- figure 3: you have to mention all the parts of the figure in the figure legend (from A to E).

11- figure 4 and topic 3.4.1. how do you explain these results inspite of different water quality parameters, you did not change the water for 7 days, with continuous feeding which will increase organic matter concentration and pH and this should affect the bacterial community from that of day 0.  

Author Response

Response to Reviewer 3 Comments:

Materials section

Point 1- line 86: please mention the name of the company and all other details between brackets.

Response 1: Thank you very much for your comments. This is how we think about the product information of a Bacillus probiotic compound. Considering the repeatability of scientific research, we are well aware that experimental reagents need to disclose product source information such as manufacturers. However, in this study, a "double blind" evaluation method was adopted, selecting a common Bacillus probiotic in the Chinese fishing probiotics market for evaluation and research, in order to objectively evaluate the probiotics and avoid suspicion of commercial interests.

In the revised manuscript, “The commercial compound Bacillus probiotic BG4 was sourced from a well-known aquatic probiotic production company in China.” was revised to “The commercial compound Bacillus probiotic BG4 was collected from a aquafarm in Guangdong province of China, which was a common aquatic probiotic product.” in line 85-86.

Point 2- line 86: how do you think the difference in bacterial concentration written on the label from the actual concentration will affect the shrimp?

Response 2: Thank you very much for your comments. Bacillus probiotics are considered to improve water quality, inhibit harmful bacteria and algal blooms, and promote the growth of shrimp and other farmed species. But the substandard quality of probiotic is likely to result in the inability to achieve water quality purification. The type and concentration of bacteria are one of the important indicators for evaluating the quality of probiotic. 

Point 3- topic 2.2.1.: activation of BG4, please include your reference for this method or if it is lab-induced method you can mention this as well.

Response 3: Thank you very much for your comments. According to reviewer's comments, the sentence were revised to “According to the method of activation of BG4 indicated on the product label, 2.5 g of BG4 and 1.5 g of brown sugar were added to an Erlenmeyer flask containing 500 mL of sterile water and placed in a shaker for 12 h at a culture temperature of 30°C and a rotation speed of 180 rpm.”

Point 4- before you add the identification step, you need to explain the culture of your bacteria. There is a missing part in your manuscript, you need to explain how you prepared bacterial culture and grown your bacteria then come to the isolation and identification step.

Response 4: Thank you very much for your comments. According to reviewer's comments, the sentence were revised to “The bacteria cultured in the nutrient agar plates were isolated and purified. And the single colonies were separated and purified on the basis of the morphological characteristics of the colonies on the counting plate.” in the revised manuscript.

Point 5- line 149: "During the experiment, the gas was inflated without changing the water." You should re-write this part as it is oxygen supply not gas.

Response 5: Thank you very much for your comments. According to reviewer's comments, the sentence were revised to “During the experiment, the water was aerated without changing the water.” in the revised manuscript.

Point 6- you repeated the experimental design twice, as under topic 2.3.3. treatment groups and topic 2.4.2. experimental design.  

The same for topic 2.3.4. and topic 2.4.3. (line 168) you repeated the same water quality measurements twice.

Again it is the same situation for " Bacterial quantity and microbial community structure " you are repeating the same paragraphs and it is not professional.

Response 6: Thank you very much for your comments. Perhaps the description of the materials and methods in the article is not clear and concise enough. These are two different experimental systems, one in a sterilized aquaculture water system and the other in a system where P. vannamei were cultured. And the bacterial quantity and microbial community structure were only analyzed in the later experiment.

According to reviewer's comments, I modified and reduced some content. “Concentrations were measured using indophenol blue spectrophotometry, naphthyl ethylenediamine spectrophotometry, zinc cadmium reduction method, phosphomolybdate blue spectrophotometry, and alkalic potassium permanganate according to GB17378.4-2007 [21].” was revised to “Concentrations were measured as previously described.” in line 168-169 of the revised manuscript.

Point 7- line 163: do not use "aquaculture organism" use the species name in your experiment.

Response 7: Thank you very much for your comments. According to reviewer's comments, “Survival rate of aquaculture organisms” was revised to “Survival rate of aquaculture Penaeus vannamei”.

Point 8- statistical analysis: the equation regarding "R" should not be placed here, instead it should be placed under water quality.

Response 8: Thank you very much for your comments. However, the calculation of degradation rate (R) is not only involved in the water quality indicators of experiment 2.3, but also in that of experiment 2.4. Therefore, I have placed all data analysis in section 2.5.

Point 9- figure1: how do you explain the differences between your treatment group (BG4) and control group in day 0? Is it logic? If there are differences in day zero, so there is something wrong with your experimental design and the downstream results are no more reliable.  

Response 9: Thank you very much for your comments. The difference between the treatment group (BG4) and the control group in day 0 mainly comes from the concentration of ammonia nitrogen and nitrite nitrogen in water. This is caused by the addition of the activation solution of Bacillus probiotic BG4 to the experimental system. Considering that the actual application of Bacillus probiotic is also after activation, the experimental design follows the practical application habits of Bacillus probiotic. The data on the 0th day of the control group can be considered as the concentration of the experimental system before adding the Bacillus probiotic, while the data on the 0th day of group BG4 is the concentration after adding the Bacillus probiotic. It is not difficult to find that the addition of bacterial activation solution to the experimental system will mainly increase the concentration of ammonia nitrogen and nitrite nitrogen in water from above data.

Point 10- figure 3: you have to mention all the parts of the figure in the figure legend (from A to E).

Response 10: Thank you very much for your comments. According to reviewer's comments, “During the experiment, the NH4+-N, NO2-N, NO3-N, PO43−-P, and COD concentrations did not significantly differ between the two groups (p>0.05; Figure 3).” was revised to “During the experiment, the NH4+-N (Figure 3A), NO2-N (Figure 3B), NO3-N (Figure 3C), PO43−-P (Figure 3D), and COD (Figure 3E) concentrations did not significantly differ between the two groups (p>0.05)”.

Point 11-figure 4 and topic 3.4.1. how do you explain these results inspite of different water quality parameters, you did not change the water for 7 days, with continuous feeding which will increase organic matter concentration and pH and this should affect the bacterial community from that of day 0.

Response 11: Thank you very much for your comments. Because the addition of probiotics only accounts for about 1% of the total number of bacteria in the water (Figure 7), the addition of probiotics changes the composition of the bacterial community structure in the water body, rather than the total number of bacteria. With the quantitative feeding twice a day, the water quality factors and bacterial community structure composition have changed (Figure 5-6), but the total number of bacteria has remained stable within 7 days.

Round 2

Reviewer 3 Report (Previous Reviewer 3)

The authors have addressed all the comments in the manuscript.

This manuscript is a resubmission of an earlier submission. The following is a list of the peer review reports and author responses from that submission.

Round 1

Reviewer 1 Report

I made numerous corrections and suggestions marked in the attached MS. Please consider them to amend your article. In general, the clarity of English composition must be improved. In detail, M&M and Discussion have to be corrected. Conclusions have to be rewritten totally.

Author Response

Response to Reviewer 1 Comments:

Point 1: Line 66-70: This part is not really relevant, should be omitted.

Response 1: Thank you very much for your comments. According to reviewer's comments, “On January 8, 2021, the Ministry of Agriculture and Rural Affairs of the People’s Republic of China issued the “Notice of the Ministry of Agriculture and Rural Affairs on Strengthening the Supervision of Aquaculture Inputs” to strengthen the supervision of aquaculture inputs, such as veterinary drugs, feed, and feed additives. Since then, the quality and degree of management of probiotics have increased to unprecedented levels.” was deleted.

Point 2: Line 86-87: It is already part of the results.

Response 2: Thank you very much for your comments. According to reviewer's comments, this sentence “The BG4 product was found to contain more than 2×108 CFU/g Bacillus, which is the main ingredient marked on its label.” has been moved from line 86-87 to line 186-187 in the revised manuscript.

Point 3: Line 130-132: What that quantity was exactly?

Response 3: Thank you very much for your comments. According to reviewer's comments, the specified bacterial quantity (5×104 CFU/mL) was added in the revised manuscript.

Point 4: Line 149-150: What does it mean? Aeration?

Response 4: Thank you very much for your comments. According to reviewer's comments, “During the experiment, the gas was inflated without changing the water.” was revised to “During the experiment, the water was inflated without changing the water.”

Point 5: Line 151: What sampling means here? Taken 120 L pond water or what?

Response 5: Thank you very much for your comments. According to reviewer's comments, “The experimental water was sampled from an intensively farmed shrimp pond.” was revised to “The experimental water came from an intensively farmed shrimp pond.”

Point 6: Line 154: If the water came from an intensively farmed pond why this adjustment was needed?

Response 6: Thank you very much for your comments. I carefully verified this experimental method, and the water used in this part of the experiment came from intensive aquaculture ponds, which were directly used in the experiment. So this part of the content has been rephrased in the revised manuscript.

Point 7: Line 193: The appropriate method of comparing two means is the T-test. However, it can be done by ANOVA, too. In this case, LSD test must be run afterwards. As it was done, I presume, so has to be mentioned.

Response 7: Thank you very much for your comments. According to reviewer's comments,“ Significant differences in the data of each group were compared through one-way ANOVA using SPSS software (version 20.0); the significance level was set at p<0.05.” was revised to “Significant differences in the data of each group were compared through one-way ANOVA after LSD test using SPSS software (version 20.0); the significance level was set at p<0.05.”.

Point 8: Line 212: The statistical significance of these differences should have been tested. Somewhere it is obvious (NO2) from the columns, somewhere (NO3) not.

Response 9: Thank you very much for your comments. According to reviewer's comments, the statistical significance of these differences are marked and supplemented in Figures 1 and topic 3.2.

Point 9: Line 331: The goal of the present study was to evaluate...

Line 332: This part in this form is unintelligible for me. What sterilization system and what bactericids?

Response 9: Thank you very much for your comments. According to reviewer's comments, “In the present study, the purpose of measuring the effect of probiotics on water quality in a sterilization system was to evaluate the effect of bactericides on water quality without the interference of other factors.” was revised to “The goal of the present study in a sterilization system was to evaluate the effect of probiotics on water quality without the interference of other factors.”

Point 10: Line 347: Which experiment? The first ot the second with shrimps?

Response 10: Thank you very much for your comments. It was the second experiment which was set in a aquaculture system where P. vannamei were cultured.

Point 11: Line 365-374: Obviously this part is the Conclusions. This is too general and what is worse these statments poorly related to your results.

It would be better to emphasize the difficulties of modelling the pond conditions in laboratory conditions and make more specific conclusions supported by your own results.

Response 11: Thank you very much for your comments. According to reviewer's comments, the conclusions were rewritten. More specific conclusions were added.

Reviewer 2 Report

This is a well written manuscript, with the experiment being well designed and executed properly and the results concisely discussion. There are a few comments on the draft manuscript the authors should clarify before it is accepted for publication, but overall this is an exceptional manuscript.

Author Response

Response to Reviewer 2 Comments:

Point 1: Line 42: "Gram" is a proper name and should start with an upper cased letter. 

Response 1: Thank you very much for your comments. According to reviewer's comments, “Bacillus is a gram-positive, spore-producing, bacterium with a high level of stress tolerance [2].” was revised to “Bacillus is a Gram-positive, spore-producing, bacterium with a high level of stress tolerance [2].”

Point 2: Line 46: “Ctenopharyngodon idellus” Please check spelling, "idella"

Response 2: Thank you very much for your comments. After verification, “Ctenopharyngodon idellus” is correct.

Point 3: Line 85 and 87: “Bacillus”  Italics?

Response 3: Thank you very much for your comments. In the revised manuscript, "Bacillus" is italicized.

Point 4: Line 97: What were these three "appropriate" dilutions???

Response 4: Thank you very much for your comments. According to reviewer's comments, “Subsequently, 0.85% sterilized normal saline was used for gradient dilution, and three appropriate dilutions were chosen.” was revised to “Subsequently, 0.85% sterilized normal saline was used for gradient dilution, and three appropriate dilutions(-5, -6, and -7 gradients) were chosen.”.

Point 5: Line 115: What was the commercial source, please list company name and location.

Line 116: This should probably be "ingredients".

Response 5: Thank you very much for your comments. According to reviewer's comments, “A commercially available compound feed containing 41% crude protein, 5% crude fat, and 16% ash as the main nutrients (mass fraction) was used in this study. ” was revised to “A commercially available compound feed (Yuehai Feed Group Co., Ltd, China) containing 41% crude protein, 5% crude fat, and 16% ash as the main ingredients (mass fraction) was used in this study. ”

Point 6: Line 125-126: This sentence needs a citation to validate the statement.

Response 6: Thank you very much for your comments. According to reviewer's comments, the reference was added in this sentence “These values were within the concentration range in water in typical aquaculture ponds [21].”

  • Cao, Y.C, Wen, G.L., Li, Z.J.,Liu,X.J., Hu, X.J., Zhang, J.S.,He, J.G. Effects of dominant microalgae species and bacterial quantity on shrimp production in the final culture season. J Appl Phycol, 2014, 26(4):1749-1757.

Point 7: Line 144: Shrimp age/growth is usually identified by their post-larval (PL) number.

Response 7: Thank you very much for your comments. According to reviewer's comments, “Similar sized (approximately 1.0 cm long) cultured P. vannamei specimens were selected.” was revised to “Similar sized (PL10) cultured P. vannamei were selected.” 

Point 8: Line 149-150: What does this mean, "gas was inflated"?

Response 8: Thank you very much for your comments. According to reviewer's comments, “During the experiment, the gas was inflated without changing the water.” was revised to “During the experiment, the water was inflated without changing the water.”

Point 9: Line 165: Replace with "shrimp". The plural of "shrimp" is "shrimp". No "s".

Response 9: Thank you very much for your comments. According to reviewer's comments, “shrimps” was revised to “shrimp”.

Point 10: Line 172-175: You could say "Concentrations were measured as previously described."

Response 10: Thank you very much for your comments. According to reviewer's comments, “Concentrations were measured using indophenol blue spectrophotometry, naphthyl ethylenediamine spectrophotometry, zinc cadmium reduction method, phosphomolybdate blue spectrophotometry, and alkalic potassium permanganate according to GB17378.4-2007[22]” was revised to “Concentrations were measured as previously described.”.

Point 11: Line 179: How is bacteria counted with a hemocytometer? This is ususally used to count and differentiate blood cells.

Response 11: Thank you very much for your comments. Hemocytometer is a commonly used cell counting tool, named after red blood cells, white blood cells, and other microorganisms in medicine. It is also commonly used to calculate the number of bacteria, fungi, yeast, and other microorganisms.

Point 12: Line 182-184:Why was the water filtered through a "filter membrane"? The next sentence says the bacteria were "extracted from water".  

Response 12: Thank you very much for your comments. According to reviewer's comments, these sentences were revised to “Water samples (100 mL) from the BG4 and control groups were filtered through a 0.22 μm filter membrane (Millipore, USA). Total bacterial DNA of water samples was extracted from these filter membrane using a microbial DNA extraction kit (Omega, USA).”

Point 13: Line 239-240:This whole figure legend is incorrect (it's the same as Figure 2 figure legend).

Response 13: Thank you very much for your comments. According to reviewer's comments, the figure legend of Figure 3 was revised to “Effect of commercial Bacillus probiotic BG4 on the water quality indicators in the shrimp culture water. (A) NH4+-N, (B) NO2--N, (C) NO3--N, (D)PO43--P, and (E) COD.

Point 14: Line 270-272:Since these are generic names, they should all start with an upper cased letter. This change includes this entire paragraph.

Response 14: Thank you very much for your comments. According to reviewer's comments, the initial letters of bacterial names have been changed to uppercase letters.

Point 15: Line 342:This should be plural "mussels".

Response 15: Thank you very much for your comments. According to reviewer's comments, “mussel” was revised to “mussels”.

Reviewer 3 Report

Materials section

1- line 86: please mention the name of the company and all other details between brackets.

2- line 86: how do you think the difference in bacterial concentration written on the label from the actual concentration will affect the shrimp?

3- topic 2.2.1.: activation of BG4, please include your reference for this method or if it is lab-induced method you can mention this as well.

4- before you add the identification step, you need to explain the culture of your bacteria. There is a missing part in your manuscript, you need to explain how you prepared bacterial culture and grown your bacteria then come to the isolation and identification step.

5- line 149: "During the experiment, the gas was 149 inflated without changing the water." You should re-write this part as it is oxygen supply not gas.

6- you repeated the experimental design twice, as under topic 2.3.3. treatment groups and topic 2.4.2. experimental design.  

The same for topic 2.3.4. and topic 2.4.3. (line 168) you repeated the same water quality measurements twice.

Again it is the same situation for " Bacterial quantity and microbial community structure " you are repeating the same paragraphs and it is not professional.

7- line 163: do not use "aquaculture organism" use the species name in your experiment.

8- statistical analysis: the equation regarding "R" should not be placed here, instead it should be placed under water quality.

9- figure1: how do you explain the differences between your treatment group (BG4) and control group in day 0? Is it logic? If there are differences in day zero, so there is something wrong with your experimental design and the downstream results are no more reliable.  

10- figure 3: you have to mention all the parts of the figure in the figure legend (from A to E).

11- figure 4 and topic 3.4.1. how do you explain these results inspite of different water quality parameters, you did not change the water for 7 days, with continuous feeding which will increase organic matter concentration and pH and this should affect the bacterial community from that of day 0.  

Author Response

Response to Reviewer 3 Comments:

Materials section

Point 1- line 86: please mention the name of the company and all other details between brackets.

Response 1: Thank you very much for your comments. When writing the manuscript, we had thought about providing specific information about  Bacillus probiotic, but some experts suggested avoiding bacterial products due to potential disputes.

Point 2- line 86: how do you think the difference in bacterial concentration written on the label from the actual concentration will affect the shrimp?

Response 2: Thank you very much for your comments. Bacillus probiotics are considered to improve water quality, inhibit harmful bacteria and algal blooms, and promote the growth of shrimp and other farmed species. But the substandard quality of probiotic is likely to result in the inability to achieve water quality purification. The type and concentration of bacteria are one of the important indicators for evaluating the quality of probiotic. 

Point 3- topic 2.2.1.: activation of BG4, please include your reference for this method or if it is lab-induced method you can mention this as well.

Response 3: Thank you very much for your comments. According to reviewer's comments, the sentence were revised to “According to the method of activation of BG4 indicated on the product label, 2.5 g of BG4 and 1.5 g of brown sugar were added to an Erlenmeyer flask containing 500 mL of sterile water and placed in a shaker for 12 h at a culture temperature of 30°C and a rotation speed of 180 rpm.”

Point 4- before you add the identification step, you need to explain the culture of your bacteria. There is a missing part in your manuscript, you need to explain how you prepared bacterial culture and grown your bacteria then come to the isolation and identification step.

Response 4: Thank you very much for your comments. According to reviewer's comments, the sentence were revised to “The bacteria cultured in the nutrient agar plates were isolated and purified. And the single colonies were separated and purified on the basis of the morphological characteristics of the colonies on the counting plate.” in the revised manuscript.

Point 5- line 149: "During the experiment, the gas was inflated without changing the water." You should re-write this part as it is oxygen supply not gas.

Response5: Thank you very much for your comments. According to reviewer's comments, the sentence were revised to “During the experiment, the water was inflated without changing the water.” in the revised manuscript.

Point 6- you repeated the experimental design twice, as under topic 2.3.3. treatment groups and topic 2.4.2. experimental design.  

The same for topic 2.3.4. and topic 2.4.3. (line 168) you repeated the same water quality measurements twice.

Again it is the same situation for " Bacterial quantity and microbial community structure " you are repeating the same paragraphs and it is not professional.

Response 6: Thank you very much for your comments. Perhaps the description of the materials and methods in the article is not clear and concise enough. These are two different experimental systems, one in a sterilized aquaculture water system and the other in a system where P. vannamei were cultured. And the bacterial quantity and microbial community structure were only analyzed in the later experiment. According to reviewer's comments, I modified and reduced some content.

Point 7- line 163: do not use "aquaculture organism" use the species name in your experiment.

Response 7: Thank you very much for your comments. According to reviewer's comments, “Survival rate of aquaculture organisms” was revised to “Survival rate of aquaculture Penaeus vannamei”.

Point 8- statistical analysis: the equation regarding "R" should not be placed here, instead it should be placed under water quality.

Response 8: Thank you very much for your comments. However, the calculation of degradation rate (R) is not only involved in the water quality indicators of experiment 2.3, but also in that of experiment 2.4. Therefore, I have placed all data analysis in section 2.5.

Point 9- figure1: how do you explain the differences between your treatment group (BG4) and control group in day 0? Is it logic? If there are differences in day zero, so there is something wrong with your experimental design and the downstream results are no more reliable.  

Response 9: Thank you very much for your comments. The difference between the treatment group (BG4) and the control group in day 0 mainly comes from the concentration of ammonia nitrogen and nitrite nitrogen in water. This is caused by the addition of the activation solution of Bacillus probiotic BG4 to the experimental system. Considering that the actual application of Bacillus probiotic is also after activation, the experimental design follows the practical application habits of Bacillus probiotic. The data on the 0th day of the control group can be considered as the concentration of the experimental system before adding the Bacillus probiotic, while the data on the 0th day of group BG4 is the concentration after adding the Bacillus probiotic. It is not difficult to find that the addition of bacterial activation solution to the experimental system will mainly increase the concentration of ammonia nitrogen and nitrite nitrogen in water from above data.

Point 10- figure 3: you have to mention all the parts of the figure in the figure legend (from A to E).

Response 10: Thank you very much for your comments. According to reviewer's comments, “During the experiment, the NH4+-N, NO2-N, NO3-N, PO43−-P, and COD concentrations did not significantly differ between the two groups (p>0.05; Figure 3).” was revised to “During the experiment, the NH4+-N (Figure 3A), NO2-N (Figure 3B), NO3-N (Figure 3C), PO43−-P (Figure 3D), and COD (Figure 3E) concentrations did not significantly differ between the two groups (p>0.05)”.

Point 11-figure 4 and topic 3.4.1. how do you explain these results inspite of different water quality parameters, you did not change the water for 7 days, with continuous feeding which will increase organic matter concentration and pH and this should affect the bacterial community from that of day 0.

Response 11: Thank you very much for your comments. Because the addition of probiotics only accounts for about 1% of the total number of bacteria in the water (Figure 7), the addition of probiotics changes the composition of the bacterial community structure in the water body, rather than the total number of bacteria. With the quantitative feeding twice a day, the water quality factors and bacterial community structure composition have changed (Figure 5-6), but the total number of bacteria has remained stable within 7 days.
